# The Dynamic Linkage between Provirus Integration Sites and the Host Functional Genome Property Alongside HIV-1 Infections Associated with Antiretroviral Therapy

**DOI:** 10.3390/vaccines11020402

**Published:** 2023-02-09

**Authors:** Heng-Chang Chen

**Affiliations:** Center for Population Diagnostics, Lukasiewicz Research Network—PORT Polish Center for Technology Development, 54-066 Wroclaw, Poland; heng-chang.chen@port.lukasiewicz.gov.pl

**Keywords:** human immunodeficiency virus type 1 (HIV-1), HIV-1 integration sites, Molecular Signatures Database (MSigBb), virus–host interaction, infection and immunity

## Abstract

(1) Background: The HIV-1 latent reservoir harboring replication-competent proviruses is the major barrier in the quest for an HIV-1 infection cure. HIV-1 infection at all stages of disease progression is associated with immune activation and dysfunctional production of proinflammatory soluble factors (cytokines and chemokines), and it is expected that during HIV-1 infection, different immune components and immune cells, in turn, participate in immune responses, subsequently activating downstream biological pathways. However, the functional interaction between HIV-1 integration and the activation of host biological pathways is presently not fully understood. (2) Methods: In this work, I used genes targeted by proviruses from published datasets to seek enriched immunologic signatures and host biological pathways alongside HIV-1 infections based on MSigDb and KEGG over-representation analysis. (3) Results: I observed that different combinations of immunologic signatures of immune cell types and proinflammatory soluble factors appeared alongside HIV-1 infections associated with antiretroviral therapy. Moreover, enriched KEGG pathways were often related to “cancer specific types”, “immune system”, “infectious disease viral”, and “signal transduction”. (4) Conclusions: The observations in this work suggest that the gene sets harboring provirus integration sites may define specific immune cells and proinflammatory soluble factors during HIV-1 infections associated with antiretroviral therapy.

## 1. Introduction

HIV-1 infection has a dynamic disease progression associated with a combination of immune cells, immune components, and various cellular restriction factors aimed at battling infections. It is known that HIV-1 infection induces a plethora of proinflammatory soluble factors that recruit and activate innate immune-related cells to the site of HIV-1 infection to restrict the replication and spread of viruses. Even though these defense mechanisms begin at the early stage of HIV-1 infections, HIV-1 still manages to complete its life cycle and establish latent reservoirs.

Several host genes, so-called recurrent integration genes [1,2,3], have been demonstrated to be frequently targeted by HIV-1 in infected individuals. However, the mechanism that leads HIV-1 integration sites to these recurrent integration genes and the biological meaning of this phenomenon remains unclear. Nevertheless, a remarkable study attempted to answer this question by showing the interaction between the selection of HIV-1 integration sites and functional pathways assigned by different biological processes in the host cells [4]. Their findings implied that the possible mechanism leading to HIV-1 integration in hot spot genes can happen at the level of the functional genome property formed by a group of genes rather than a gene per se.

In the continuation of this concept, in this work, I sought any functional linkage between provirus-targeted genes and host immunity alongside HIV-1 infections. I used the host genes targeted by proviruses [5,6] as an input gene list. This list of genes included integration sites retrieved from elite controllers [6] and HIV-1-infected individuals subjected to antiretroviral therapy (ART) [5]. Of note, the integration sites published by Einkauf et al. (2022) [5] were longitudinally collected from HIV-1-infected individuals, thereby allowing us to follow how the HIV-1 reservoir evolves during the period of ART. Based on the over-representation analysis, I observed that different immunologic signatures were enriched alongside HIV-1 infections associated with ART. In contrast, few immunologic signatures were enriched in elite controllers. In addition, enriched gene sets were mainly involved in biological pathways related to “cancer specific types”, “immune system”, “infectious disease viral”, and “signal transduction”. Finally, I found that enriched immunologic signatures were contributed by provirus-targeted genes that are associated with gene products of HIV-1 or affect HIV-1 replication and infectivity. Altogether, these findings appeared to resemble the concept of network theory [7,8,9] that provirus-targeted genes engaged in similar biological functions may form a community (refer to a “signature” in this study) to satisfy the need for host immunity during different periods of HIV-1 infections associated with ART. Based on this logic, I proposed a hypothesis that there could be a dynamic interplay between HIV-1 integration sites and the host functional genome property: HIV-1 integration frequency might be used as a surrogate for gene sets, which may define specific immune cell types and proinflammatory soluble factors during HIV-1 infection.

## 2. Materials and Methods

### 2.1. Acquisition and Procession of Public Datasets

HIV-1 integration sites were collected from Jiang et al. (2020) [6] for elite controllers and Einkauf et al. (2022) [5] for pretreatment HIV-1-infected individuals (viremic) and patients subjected to a short (1 year) and a long period (7.5–12 years) of ART. A total of 232 integration sites from 64 elite controllers applied by Jiang et al. (2020) [6] were analyzed in this study. Among them, 103 sites were inserted by intact proviruses. Integration sites retrieved from elite controllers were overlaid on Human_genome_GENCODE_v32.bed released by the GENCODE project [10] to determine if the insertion of each HIV-1 integration is genic or non-genic by executing the command intersect with default options provided in bedtools [11]. Since multiple proviruses were observed to integrate into the same genomic position, in the end, 104 provirus-targeted genes were released; among them, 22 genes were targeted by intact proviruses. A total of 1270 provirus-targeted genes from 6 patients (P1: 468 integration sites; P2: 149 integration sites; P3: 128 integration sites; P4: 94 integration sites; P5: 305 integration sites; P6: 125 integration sites) used in Einkauf et al. (2022) [5] were selected for this study. In total, 317 integration sites that fall into the category of pretreatment HIV-1-infected individuals (viremic) yielded 121 provirus-targeted genes, 396 integration sites that fall into the category of patients subjected to a short period (1 year) of ART yielded 149 provirus-targeted genes, and 557 integration sites that fall into the category of patients subjected to a long period (7.5–12 years) of ART yielded 176 provirus-targeted genes. Among them, 26 intact provirus-targeted genes were found in pretreatment HIV-1-infected individuals, 6 intact provirus-targeted genes were found in patients subjected to a short period of ART, and 14 intact provirus-targeted genes were found in patients subjected to a long period of ART. Lists of the input genes used for the analysis are shown in Table 1; the genes targeted by intact proviruses are summarized in Appendix A.

### 2.2. Data and Bioinformatic Analyses

#### 2.2.1. MsigDb Over-Representation Analysis

Enriched immunologic signatures were computed using the R package clusterProfiler (Version 4.4.1) [12,13] with the function enricher and default options. Over-representation analysis [14] was performed using C7 immunologic signature gene sets curated in the Molecular Signatures Database (MsigDb) [15,16,17] as a reference. Only the enriched immunologic signatures with *p*-values (adjusted using the Benjamini–Hochberg method) of less than 0.05 were selected. Readouts of over-representation analyses generated in this study were all curated in Appendix A. Rich factors [13] were further calculated to represent the enrichment score for every enriched immunologic signature (Figure 1c) by dividing GeneRatio by BgRatio with the command lines described below.


>MsigDb_output_file$GeneRatio <-**as.numeric**(**gsub**(“(\\d+)/(\\d+)”, “\\1”, MsigDb_output_file$GeneRatio, perl = T))/**as.numeric**(**gsub**(“(\\d+)/(\\d+)”, “\\2”, MsigDb_output_file$GeneRatio, perl = T))



*# Convert GeneRatio to numerical variables.*



>MsigDb_output_file$BgRatio <-**as.numeric**(**gsub**(“(\\d+)/(\\d+)”, “\\1”, MsigDb_output_file$BgRatio, perl = T))/**as.numeric**(g**sub**(“(\\d+)/(\\d+)”, “\\2”, MsigDb_output_file$BgRatio, perl = T))



*# Convert BgRatio to numerical variables.*



>MsigDb_output_file <-MsigDb_output_file %>% **dplyr::mutate**(rich_factor = GeneRatio/BgRatio)



*# Calculate rich factors.*


**Figure 1 vaccines-11-00402-f001:**
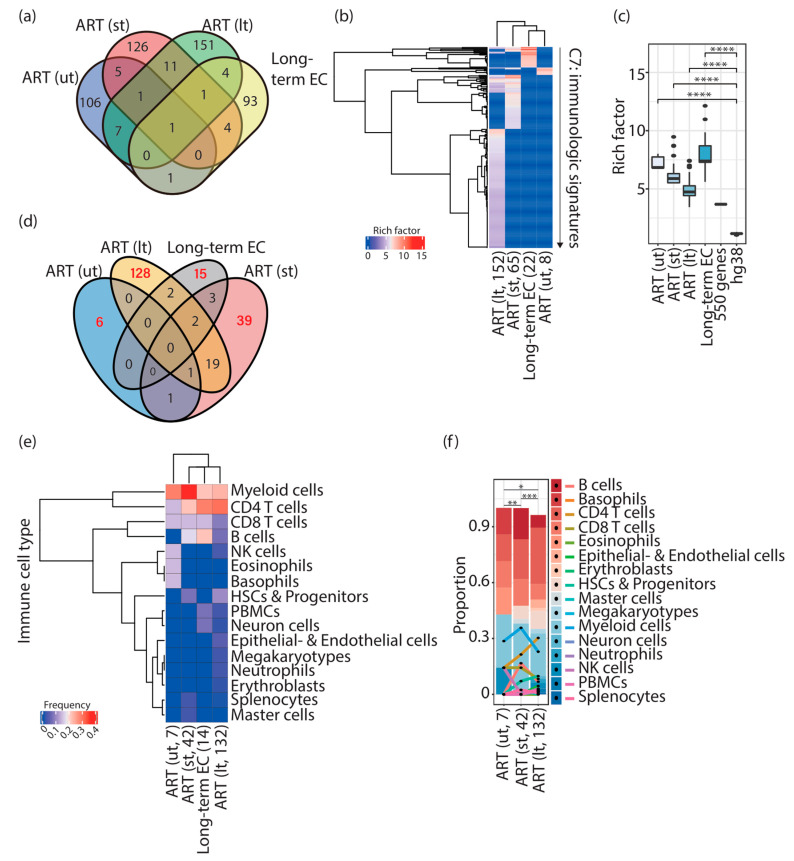
Distinct immunologic signatures were enriched in HIV-1-infected individuals and elite controllers. (**a**) Venn diagram representing the overlap of input genes in HIV-1-infected individuals and elite controllers. Genes shared among different groups of input genes are listed in Appendix A. (**b**) Cluster heatmap representing immunologic signatures enriched in HIV-1-infected individuals and elite controllers. Parentheses placed under the column indicate the status of ART (“lt” refers to HIV-1-infected individuals subjected to a long period of ART; “st” refers to HIV-1-infected individuals subjected to a short period of ART; “ut” refers to pretreatment HIV-1-infected individuals) followed by the quantity of enriched immunologic signatures. The color scale represents the magnitude of the enrichment represented by rich factors. (**c**) Box plots representing the enrichment of immunologic signatures represented by rich factors. Commands executed to calculate rich factors are described in the Materials and Methods. Statistical significance was calculated using the Wilcoxon test in R with default options. (**d**) Venn diagram representing the overlap of enriched immunologic signatures in HIV-1-infected individuals and elite controllers. Red numbers indicate the quantity of unique enriched immunologic signatures. (**e**) Cluster heatmap representing the frequency of immune cell types present in unique enriched immunologic signatures in HIV-1-infected individuals and elite controllers. Parentheses placed under the column indicate the number of immune cell types counted in the enriched immunologic signatures. (**f**) Stacked bar chart representing the proportion of the different compositions of immune cell types in pretreatment HIV-1-infected individuals and patients receiving ART. Parentheses placed under the column indicate the number of immune cell types counted in the enriched immunologic signatures. Statistical significance was calculated using Pearson’s chi-square test in R with default options, * *p* ≤ 0.05, ** *p* ≤ 0.01, *** *p* ≤ 0.001, **** *p* ≤ 0.0001. ART; antiretroviral therapy. EC; elite controllers. HSCs; hematopoietic stem cells. NK cells; natural killer cells. PBMCs; peripheral blood mononuclear cells.

Notably, after performing the MsigDB over-representation analysis, terms of immune cell types and proinflammatory soluble factors shown in the standard names of the enriched immunologic signatures were collected for further statistical calculation. Based on the Blood Cell Lineage chart published in the NIH website’s SEER Training Modules (https://training.seer.cancer.gov/leukemia/anatomy/lineage.html (accessed on 10 September 2022)), immune cell types were classified into 16 groups, namely, (1) B cells, including plasma cells, (2) basophils, (3) CD4 T cells, (4) CD8 T cells, (5) eosinophils, (6) epithelial and endothelial cells, (7) erythroblasts, (8) hematopoietic stem cells (HSCs) and progenitors, (9) master cells, (10) megakaryocytes, (11) myeloid cells, (12) neuron cells, including microglia, (13) neutrophils, (14) Natural Killer (NK) cells, (15) peripheral blood mononuclear cells (PBMCs), and (16) splenocytes. Group (8) hematopoietic stem cells (HSCs) and progenitors included stem cells and thymocytes; group (11) myeloid cells included bone-marrow-derived dendritic cells, bone-marrow-derived macrophages, dendritic cells, monocyte-derived macrophages, and macrophages. The percentages or abundances of signatures of immune cell types and proinflammatory soluble factors from the total number of enriched immunologic signatures were represented as cluster heatmaps generated using the R package ComplexHeatmap [18] with default options. All the plots presented in this manuscript were generated in R with default options.

To obtain immunologic signatures enriched by randomly selected genes and the whole genome, protein-coding genes in humans were retrieved from Human_genome_GENCODE_v32.bed released by the GENCODE project [10], and the R command sample_n() (with replacement) was executed to randomly select 150, 250, 350, 450, 550, 650, and 1000 human genes, or all protein-coding genes were used to perform the MSigDb over-representation analysis described above. 

#### 2.2.2. KEGG Pathway Over-Representation Analysis

Enriched KEGG pathways were computed by conducting KEGG pathway over-representation analysis using the R package clusterProfiler [12] with the *p*-value cutoff equal to 0.5. Based on KEGG BRITE hierarchies documented at https://www.genome.jp/kegg/brite.html, accessed on 2 October 2022, every enriched pathway was assigned to its corresponding KEGG BRITE classification to generate the plots in Figure 2b–d. Of note, to create Figure 2c, enriched KEGG pathways that were found in all group III factors (so-called “Shared hsaID” in the figure panel) were first separated from those that were only present in individual group III factors, and then the cluster heatmap was created.

#### 2.2.3. Determination of Provirus-Targeted Genes Interacting with HIV-1

The HIV-1 Human Interaction Database (https://www.ncbi.nlm.nih.gov/genome/viruses/retroviruses/hiv-1/interactions/, accessed on 30 October 2022) was used to verify whether provirus-targeted genes interact with HIV-1 or not. This database documents all known interactions of HIV-1 and human genes that have been reported to affect HIV-1 replication and infectivity as well as interactions between HIV-1 gene products and host cell proteins, or proteins from disease organisms associated with HIV-1/AIDS [19,20,21].

#### 2.2.4. Statistics

All statistical tests were performed in R with default options. Details are provided where appropriate in the main text.

## 3. Results

### 3.1. Different Immunologic Signatures Were Enriched Alongside HIV-1 Infections Associated with ART

In total, 121, 149, and 176 genes from pretreatment HIV-1-infected individuals (ut), HIV-1-infected individuals subjected to a short period of ART (st), and those subjected to a long period of ART (lt) [5], respectively, and 104 genes from elite controllers [6] (Table 1) were employed to perform over-representation analysis. A few of the input genes were shared among ut, st, lt, and elite controllers (Figure 1a and Appendix A).

In total, 8 (Appendix A), 65 (Appendix A), 152 (Appendix A), and 22 (Appendix A) immunologic signatures were enriched in pretreatment HIV-1-infected individuals, HIV-1-infected individuals subjected to a short period of ART, those subjected to a long period of ART, and elite controllers, respectively (Figure 1b). It is essential to stress that due to a lack of expression data corresponding to integration sites used in this work, I therefore calculated rich factors [13] (see Materials and Methods), rather than net enrichment scores, to represent the enrichment intensity. I used 150, 250, 350, 450, 550, 650, and 1000 randomly selected genes (data not shown) and the whole genome as controls in the over-representation analysis to validate the significance of the score of the rich factor (Figure 1c). None of the immunologic signatures was enriched using 150, 250, 350, 450, 650, and 1000 genes (data not shown), and one immunologic signature was enriched using 550 randomly selected genes (rich factor: 3.681). Although 4872 immunologic signatures were enriched using the whole genome (rich factor: median, 1.162; mean, 1.154), rich factors measured in HIV-1-infected individuals (both pretreatment and receiving ART) and in elite controllers showed significance compared to controls (Figure 1c, controls: 550 genes and hg38 labeled in the figure panel), indicating that the immunologic signatures detected in this study were significantly enriched.

### 3.2. Different Combinations of Immune Cell Signatures Were Observed Alongside HIV-1 Infections Associated with ART

In line with the finding that provirus-targeted genes enriched specific immunologic signatures alongside HIV-1 infections associated with ART, I continuously investigated whether these enriched signatures could assign specific immune cell types and proinflammatory soluble factors that are stated in the hypothesis of this work. I retrieved immunologic signatures uniquely enriched in pretreatment HIV-1-infected individuals (ut, 6 signatures, 75%), HIV-1-infected individuals subjected to a short period of ART (st, 39 signatures, 60%), HIV-1-infected individuals subjected to a long period of ART (lt, 128 signatures, 84%), and elite controllers (15 signatures, 68%) (Figure 1d). From them, I counted the appearance of names of immune cell types in the standard names of the enriched immunologic signatures one by one (Figure 1e,f). I further classified immune cell types into 16 groups based on the Blood Cell Lineage chart published in the NIH website’s SEER Training Modules (subtypes of cells in each cell category are described in the Materials and Methods). It is important to note that ART interrupts HIV-1/AIDS progression; the composition of immune cell types observed here was in physiological HIV-1 infection conditions associated with ART. First, an increased variety of immune cell types in HIV-1-infected individuals subjected to a long period of ART (lt, found 132 terms of immune cells) were found (Figure 1e,f); among them, CD4 T cells (found 40 times, 30.3%) showed the highest frequency, followed by myeloid cells (found 35 times, 26.5%) (Figure 1e,f). This composition of immune cell signatures showed a difference between pretreatment HIV-1-infected individuals (ut, found 7 times) and patients subjected to a short period of ART (st, found 42 times) (Figure 1e,f, Pearson’s chi-square test). The composition of the immune cell signatures in elite controllers (found 14 times) was unique in pretreatment HIV-1-infected individuals and ART-treated patients (Figure 1e). Of note, the proportion of the group “B cells” was highly enriched in elite controllers compared to ART-treated patients (found 3 times, 21.4%) (Figure 1e); whereas, in ART-treated patients, it showed a decrease in HIV-1-infected individuals subjected to a short (found 7 times, 16.7%) and a long period of ART (found 9 times, 6.8%) (Figure 1e,f). The group “B cells” was not found in unique enriched immunologic signatures in pretreatment HIV-1-infected individuals (Figure 1e,f). CD4 T cells, the main HIV-1 reservoir cell type, showed an increase in their proportion alongside HIV-1 infections associated with ART (ut, found 1 time, 14.3%; st, found 9 times, 21.4%; lt, found 40 times, 30.3%). A regression of the presence of CD8 T cells (ut, found 1 time, 14.3%; st, found 6 times, 14.3%; lt, found 11 times, 8.3%) was, however, observed after a long period of ART (Figure 1e,f). The highest peak of the proportion of the myeloid cell signature was found in HIV-1-infected individuals subjected to a short period of ART (found 15 times, 35.7%). Additionally, the proportion of myeloid cells between pretreatment HIV-1-infected individuals (found 2 times, 28.6%) and patients subjected to a long period of ART (found 35 times, 26.5%) was comparable (Figure 1f). Intriguingly, the presence of the NK cell signature was observed in pretreatment HIV-1-infected individuals (found 1 term, 14.3%) and HIV-1-infected individuals subjected to a long period of ART (found 4 times, 3%) (Figure 1f). Collectively, these findings suggest that different combinations of immune cells with an increasing variety were present alongside HIV-1 infections with ART.

### 3.3. Different Combinations of Proinflammatory Soluble Factor Signatures Were Observed Alongside HIV-1 Infections Associated with ART

In the continuation of the previous logic, I also counted the appearances of the names of proinflammatory soluble factors, including cytokines and chemokines, in each unique enriched immunologic signature (Figure 2a), and found that 1, 8, 33, and 6 unique immunologic signatures contained the names of proinflammatory soluble factors in pretreatment HIV-1-infected individuals, HIV-1-infected individuals subjected to a short period of ART, those subjected to a long period of ART, and elite controllers, respectively (Figure 2a). I further grouped these proinflammatory soluble factors based on their appearance in HIV-1-infected individuals: group I factors, including IL4 and IL12, were present in pretreatment HIV-1-infected individuals and throughout the whole infection process (Figure 2a); group II factors, including CXCR5, IFNG, IFNB, IL10, and TGFB, were present in HIV-1-infected individuals subjected to ART irrespective of the period (Figure 2a); and group III factors, including CXCL4, IFNA, IL1, IL2, IL6, IL7, IL15, IL18, and TNF, appeared only after a long period of ART (Figure 2a). To further investigate each group of proinflammatory soluble factors, I sought KEGG pathways enriched by the genes retrieved from unique enriched immunologic signatures with corresponding proinflammatory soluble factors (Figure 2b,c). No clear separation of enriched KEGG pathways related to group II factors between HIV-1-infected individuals subjected to a short and a long period of ART was observed (Figure 2b), implying that the enrichment of these pathways was irrelevant to the period of therapy. Nevertheless, enriched pathways in “cancer specific types” were often associated with IFNB (lt), CXCR5 (st), and IFNG (lt); enriched pathways in “signal transduction” and “immune system” were more related to IL10 (st) and IFNB (st); enriched pathways in “infectious disease viral” were more related to CXCR5 (st) (Figure 2b). As for group III factors, pathways in “signal transduction”, “immune system”, and “infectious disease viral” were prevalent for every proinflammatory soluble factor in this group (Figure 2c, the first column from the left). Pathways in “membrane transport” (hsaID uniq. IL12), “glycan biosynthesis and metabolism” (hsaID uniq. IL17), “signaling molecules and interaction” (hsaID uniq. IL1), “carbohydrate metabolism” (hsaID uniq. CXCL4), “replication and repair” (hsaID uniq. CXCL4), and “endocrine and metabolic disease” (hsaID uniq. CXCL4) became specific to the indicated proinflammatory soluble factors (Figure 2c). As I compared the enriched KEGG pathways between group II and group III factors, more than half (61 pathways) of the pathways were common (Figure 2d). Among them, “infectious disease viral”, “immune system”, and “signal transduction” were the top three KEGG BRITE classifications shared in both groups of factors (Figure 2e). Enriched pathways in the classification “cancer specific types” were only found in group II factors (Figure 2e). Collectively, these findings suggest that different combinations of proinflammatory soluble factors participated in immunity alongside HIV-1 infections associated with ART.

To verify whether enriched pathways involved in “cancer specific types”, “immune system”, “signal transduction”, and “infectious disease viral” were truly required during HIV-1 infections, I withdrew all genes present in these four KEGG BRITE classifications from the input gene list (Table 1) and repeated the over-representation analysis performed in Figure 1b. First, I observed that the majority of genes present in “cancer specific types”, “immune system”, and “infectious disease viral” are common, whereas the genes present in “signal transduction” appeared to be distinct from the other three classifications (Figure 3a). Intriguingly, genes present in these four KEGG BRITE classifications were restricted to HIV-1-infected individuals receiving ART and specific to different periods of treatment (Figure 3b–e), except the genes *rapgfe2* and *ppp3cc.* The gene *rapgfe2* was observed throughout the whole period of HIV-1 infections (Figure 3e), and *ppp3cc* appeared after patients received ART (Figure 3c–e). The gene *cyth1* was found in pretreatment HIV-1-infected individuals and patients subjected to a long period of ART; whether the absence of *cyth1* in HIV-1-infected individuals subjected to a short period of ART (Figure 3e) is due to technical reasons or is biologically relevant will require further investigation. Furthermore, a decreased number of enriched immunologic signatures with minor significance were obtained as genes in these four KEGG BRITE classifications, and were withdrawn (Figure 3f–g) (Appendix A). Additionally, in HIV-1-infected individuals subjected to a long period of ART, several proinflammatory soluble factor signatures (IL4, IL6, IL7, IL15, IFNG) were infrequently present (Figure 3h), especially when the genes involved in “signal transduction” were taken away (Figure 3h, column 5). Consistently, the same pattern was present in the correlation plot as well (Figure 3j). However, signatures related to IL3, IL12, and IL18 were not affected (Figure 3h). Other signatures appeared to be either random (IFNA) or slightly affected (CXCL4, CXCR5, IFNB, IL1, IL2, IL10, IL35, TGFB) when genes in different KEGG BRITE classifications were withdrawn (Figure 3h, columns 1–5). The impact of the genes retrieved from HIV-1-infected individuals subjected to a short period of ART was minor (Figure 3h), thereby causing a clear separation of two main clusters between short and long periods of ART (Figure 3h). Nevertheless, a decreased correlation was still observed in HIV-1-infected individuals subjected to a short period of ART when genes in “immune system” were taken away (Figure 3i).

### 3.4. The Majority of Enriched Immunologic Signatures Resulted from Gene Sets That Interact with HIV-1

As I hypothesized that HIV-1 integration frequency might be used as a surrogate for gene sets to define specific immune cell types and proinflammatory soluble factors required during HIV-1 infections, I therefore tested whether any connection is present between HIV-1 integration and the targeted genes. If this is the case, I expect that the required immunologic signatures are enriched by the gene sets that either show protein interactions with gene products of HIV-1 or affect HIV-1 replication and infectivity; whereas, other gene sets that do not interact with HIV-1 are not responsible for the enrichment of any immunologic signature during HIV-1 infections. To verify this hypothesis, I separated input genes reported to interact with HIV-1 from others that do not interact with HIV-1 (see Materials and Methods), and observed that 38.8% (47 genes out of 121 genes), 40.5% (60 genes out of 148 genes), 37.5% (66 genes out of 176 genes), and 29.8% (31 genes out of 104 genes) of the genes have been reported to interact with HIV-1 in pretreatment HIV-1-infected individuals, HIV-1-infected individuals subjected to a short period of ART, those subjected to a long period of ART, and elite controllers, respectively. The majority of the genes showed interactions with HIV-1 *gag-pol* genes, followed by the *tat* gene, the *env* gene, and the *nef* gene; few HIV-1-targeted genes were found to interact with the HIV-1 *vpr*, *vif*, *rev*, and *vpu* genes (Figure 4a).

I further performed over-representation analysis on both subsets of input genes (interact with HIV-1 or not) and found that the enrichment of the immunologic signatures was mainly contributed by the genes interacting with HIV-1 (Figure 4b) (Appendix A). Only a tiny proportion of immunologic signatures were enriched by the genes that do not interact with HIV-1 (Appendix A) (Figure 4b). Notably, genes interacting with HIV-1 in pretreatment HIV-1-infected individuals were only responsible for a small proportion of the enriched immunologic signatures (Figure 4b) (Appendix A). None of the immunologic signatures can be enriched using the genes that do not interact with HIV-1 in pretreatment HIV-1-infected individuals (Figure 4b). Intriguingly, rich factors showed an increase in HIV-1-infected individuals subjected to ART, regardless of whether provirus-targeted genes were reported to interact with HIV-1 or not (Figure 4c). Furthermore, a variety of immune cell signatures were only observed using genes interacting with HIV-1 in patients receiving a short (145 terms) and a long period (59 terms) of ART (Figure 4d), resembling similar circumstances using a complete list of input genes. In contrast, a poor combination of immune cell types was found in enriched immunologic signatures using genes that do not interact with HIV-1 (Figure 4d). Consistently, the majority of the proinflammatory soluble factor signatures were present in enriched immunologic signatures using a complete list of input genes and the genes interacting with HIV-1 in infected individuals subjected to ART (Figure 4e). Either a tiny proportion of proinflammatory soluble factor signatures or no proinflammatory soluble factor signatures were observed using the genes that do not interact with HIV-1 and using the genes retrieved from pretreatment HIV-1-infected individuals (Figure 4e). Altogether, these findings indicate that enriched immunologic signatures in HIV-1-infected individuals subjected to ART mainly resulted from gene sets interacting with HIV-1.

## 4. Discussion

In this work, I focused on the possible interplay between host genes targeted by proviruses and the host functional genome property. Based on the over-representation analysis of the genes targeted by proviruses, I first observed that different unique immunologic signatures were enriched alongside HIV-1 infections associated with ART (Figure 1b). Furthermore, an increase in the variation in immune cell types was present in HIV-1-infected individuals subjected to a long period of ART (Figure 1e,f), reflecting an environment with complicated immune responses. Given that different kinds of innate immune-related cells are recruited to the site of HIV-1 infection over time, unveiling the alterations of immune cells alongside HIV-1 infections can thus be essential to understand the evolution of HIV-1 reservoirs. As shown in this work, different compositions of immune cell types were present in unique enriched immunologic signatures in pretreatment HIV-1-infected individuals, ART-treated HIV-1-infected patients, and elite controllers (Figure 1e,f). This observation can also be linked to different compositions of proinflammatory soluble factors alongside HIV-1 infections associated with ART, as observed in this work (Figure 2a).

This study demonstrated that different combinations of proinflammatory soluble factor signatures were present at different stages of ART. I observed that CXCR5, IFNB, IFNG, IL10, and TGFB were found in enriched immunologic signatures in HIV-1-infected individuals subjected to a short and a long period of ART (Figure 2a). CXCL4, IFNA, IL1, IL2, IL6, IL7, IL15, IL18, and TNF were observed only in enriched immunologic signatures in HIV-1-infected individuals subjected to a long period of ART (Figure 2a). Few proinflammatory soluble factors were observed in pretreatment HIV-1-infected individuals and elite controllers (Figure 2a). Indeed, it has been known for a long time that HIV-1 infections result in dysregulation of the cytokine profile [22]. Further investigations will be required to understand the immune strategy using different compositions of proinflammatory soluble factors to interfere with the evolution of an HIV-1 latent reservoir. KEGG pathways enriched by gene sets related to group II and III factors mainly fell into the KEGG BRITE classifications “cancer-specific types”, “infectious disease viral”, “immune system”, and “signal transduction” (Figure 2b–e). It is essential to note that gene sets from these four KEGG BRITE classifications were restricted to HIV-1-infected individuals subjected to a short and a long period of ART, and the majority of them were specific to different periods of treatment, indicating that the selection of proviral integration can be disease progression-specific (Figure 3b–e). According to the observation in this work, this selection is irrelevant to elite control (Figure 3b–e). Finally, I showed that the enriched immunologic signatures were mainly contributed by genes interacting with HIV-1 (Figure 4b), even though the genes reported to interact with HIV-1 did not represent the majority of the genes in the input list. Here, it is intriguing to point out that these founder genes analyzed in this work not only showed interaction with HIV-1 but also were invaded by the genome of the virus in physiological HIV-1 infection conditions. This observation may indicate that the affinity of the provirus-targeted genes towards gene products of HIV-1 can serve as a potential indicator to predict the required immunologic signatures during HIV-1 infections associated with ART. The following are open questions: What makes the genes reported to interact with HIV-1 different from others? What is the role of the provirus-targeted genes that do not interact with HIV-1? 

Although it is presently workable to sequence the nearly full-length HIV-1 genome, the yield of retrieved sequences is limited to the range of 50 to 100 sequences in cells purified from a patient [23,24], thereby rendering accurate and quantitative investigation difficult. A better strategy to sequence the full-length HIV-1 genome of proviruses and elevate the yield of output sequences from clinical samples will be an absolute requisite because intact proviruses are responsible for viral rebounds after treatment cessation [24]. In total, 26, 6, 14, and 22 input genes were targeted by intact proviruses in pretreatment HIV-1-infected individuals, patients subjected to a short period of ART, those subjected to a long period of ART, and elite controllers, respectively (Appendix A). Intriguingly, around one third to half of the intact provirus-targeted genes were associated with the signatures of immune cell types (10 genes in pretreatment HIV-1-infected individuals, 3 genes in patients subjected to a short period of ART, 7 genes in patients subjected to a long period of ART, and 11 genes in elite controllers) (Appendix A). Conversely, intact provirus-targeted genes were not frequently associated with the signatures of proinflammatory soluble factors (2 genes in pretreatment HIV-1-infected individuals, 1 gene in patients subjected to a short period of ART, 4 genes in patients subjected to a long period of ART, and 4 genes in elite controllers) (Appendix A). A more recent study published by Sun et al. (2023) [25] continued to strengthen this “immune selection hypothesis” proposed by Einkauf and colleagues (the term “immune selection hypothesis” was mentioned in Peer Review File available at https://static-content.springer.com/esm/art%3A10.1038%2Fs41586-022-05538-8/MediaObjects/41586_2022_5538_MOESM2_ESM.pdf (accessed on 25 January 2023)). They observed that HIV-1 reservoir cells harboring intact proviruses frequently express ensemble phenotypic signatures of surface markers (including the checkpoint marker PD-1) that confer resistance to immune-mediated killing by CD8+ T cells and NK cells [25]. In this study, two unique immunologic signatures, GSE26495_NAIVE_VS_PD1LOW_CD8_TCELL_DN [26] and GSE24026_PD1_LIGATION_VS_CTRL_IN_ACT_TCELL_LINE_DN [27], enriched in HIV-1-infected individuals subjected to a long period of ART were observed to be associated with PD-1 (Appendix A).

In this work, I investigated whether the functional genome property also contributes to this immune selection alongside HIV-1 infections associated with ART and tested if genes targeted by proviruses can form different communities that are beneficial to the immunological process by recruiting immune cells and activating signaling pathways in different pathogenic stages of HIV-1 infections associated with ART. It is important to note that although the enriched immunologic signature gene sets represented less than 10% of the input genes (Appendix A), the rich factors showed significant enrichment (Figure 1c and Figure 4c), making the results from the over-representation analysis robust. A previous study also demonstrated that over-representation of pathways is not correlated with the number of genes in the pathways [28]. Several important questions still need to be addressed regarding (1) whether host genes associated with innate immunity are more frequently targeted by proviruses, and (2) whether HIV-1 integration in genes relevant to innate immunity is due to a transcriptionally active status, thereby being favorable for HIV-1 integration. Among the input genes analyzed in this study, very few genes overlapped between ART-treated patients and elite controllers, and between pretreatment HIV-1-infected individuals and patients subjected to ART (Figure 1a and Appendix A). The gene *rptor* is the only gene present in the four lists of input genes (Appendix A). The gene product of *rptor* forms a stoichiometric complex with the mammalian target of rapamycin (mTOR) kinase. mTOR signaling largely regulates cell metabolism in activated T cells and is known to be required for HIV-1 replication and latency [29,30], highlighting the importance of immunometabolism involved in HIV-1 pathogenesis [31]. The gene *rapgef2* was found throughout the infection course of HIV-1 in patients (Figure 3e and Appendix A). The gene product of *rapgef2* is responsible for RAS activation [32]. The Ras protein serves as a central intermediate in many signaling pathways, including NF-κB activation [33]. NF-κB is one of the critical transcription factors required for the initiation of HIV-1 transcription [34,35] and a therapeutic target activated by antiretroviral drugs [36]. Five common genes were found in pretreatment HIV-1-infected individuals and patients subjected to a short period of ART; however, little is known about their roles in HIV-1 pathogenesis (Appendix A). Of note, the function of the gene product of *fanca* is involved in DNA repair. Recently, Kabi and Filion proposed their initial hypothesis that DNA repair may help flag integrated viruses, including HIV-1, allowing the host immunity to sense viral DNA [37]. If this hypothesis is tested to be true, it would open a new avenue of research on integrated viral immune sensing, where chromatin would play a role. Seven genes are identical between pretreatment HIV-1-infected individuals and patients subjected to a long period of ART (Appendix A). The gene *malat1*, which produces a long non-coding RNA, has been shown to promote HIV-1 transcription and infection [38]. The gene product of *vav1* and its mediated signaling pathway has been shown to interact with the HIV-1 Nef [39,40,41]. A total of 11 genes were shared between patients subjected to a short and a long period of ART (Appendix A). Among them, the gene *brd4* was observed. BRD4 is one of the essential epigenetic readers that regulate HIV-1 transcription [42,43,44] and has become a popular target in the design of antiretroviral drugs [45]. In one research article, the influence of *ndufa6* gene expression on HIV-1-induced T-cell apoptosis was found [46]. Here, I have summarized input genes that are associated with signatures of either immune cell types or proinflammatory soluble factors in Table 2. 

Intriguingly, more than 90% of the input genes were associated with immune cell type signatures in HIV-1-infected individuals subjected to a long period of ART, followed by HIV-1-infected individuals subjected to a short period of ART (67.7%), elite controllers (46.1%), and pretreatment HIV-1-infected individuals (28.9%) (Table 2). Consistently, 61.9% of the input genes were associated with the signatures of proinflammatory soluble factors in HIV-1-infected individuals subjected to a long period of ART, followed by HIV-1-infected individuals subjected to a short period of ART (24.8%), elite controllers (24%), and pretreatment HIV-1-infected individuals (5.7%) (Table 2). While overlapping with human cytokine genes (in total 1793 unique genes) available at ImmPort (https://www.immport.org (accessed on 27 January 2023)) [47,48,49], none of the provirus-targeted genes associated with proinflammatory soluble factors in pretreatment HIV-1-infected individuals belonged to cytokine genes. Five (*ctss*, *nr2c2*, *pik3r1*, *prkcq*, *rac2*), ten (*bach2*, *eed*, *grap2*, *il27ra*, *il2rb*, *itga*, *jak1*, *micb*, *ppp3cc*, *tap2*) and one (*grb2*) provirus-targeted genes associated with proinflammatory soluble factors belong to cytokine genes in st, lt, and elite controllers, respectively. The gene *pik3r1* has been proposed to be one of the essential host factors targeted by HIV-1 viral proteins using genome-wide RNAi screening [50]. Several studies have shown that protein kinase C theta, the gene product of *prkcq*, is involved in controlling HIV-1 replication [51,52] and is associated with HIV-1 Nef which leads to activation of ERK1/2 [53,54]. The gene product of *rac2* (Rac) and its relevant signaling pathway has been reported to be activated by HIV-1 Nef-associated protein complex [55] and involved in HIV-1 Nef-induced blockade of superoxide release in human macrophages [56]. The gene *bach2* is presently one of the well-known cancer-related genes that are favorable for HIV-1 integration identified in HIV-1-infected individuals [1]. The gene *eed* encodes a member of the polycomb complex 2 that interacts with the HIV-1 matrix protein. Inhibition of EED allows HIV-1 latency reversal by decreasing the level of H3K27 tri-methylation [57]. The gene *il17ra* encodes interleukin 27 receptor subunit alpha. Recently, IL-27 signaling (IL-27 and its receptor) has been proposed to be involved in the pathogenesis of HIV-1 infection and immune reconstitution in infected individuals subjected to ART [58,59]. The gene *il2rb* encodes interleukin 2 receptor subunit beta. Expression of IL2RB required for the production of granulysin used by lymphocytes to kill tumors and microbial cells has been shown to be dysfunctional in HIV-1-infected individuals [60]. The gene *jak1* is a key component of the interleukin-6/JAK1/STAT3 immune and inflammation response. JAK1/2 inhibitors, such as ruxolitinib and baricitinib, have been approved by U.S. Food and Drug Administration (FDA) as a potential therapeutic strategy to target the HIV-1 latent reservoir [61]. The gene *micb* encodes a heavily glycosylated protein which is a ligand for the natural killer group 2 member type II receptor; this receptor is responsible for host immunosurveillance and NK cell-mediated cytotoxicity during HIV-1 infections [62]. The gene *tap2* is one of the MHC-II-encoded genes necessary for the generation of a cellular immune response. One study has shown that the heterozygous A/G and homozygous G/G polymorphisms on TAP2, rather than TAP1, were positively associated with an increased risk of HIV-1/AIDS infection [63]. Rom and colleagues (2011) have shown the interaction between the cellular factor Grb2 (the gene product of *grb2*) and the HIV-1 Tat protein subsequently impairs the activation of the Raf/MAPK pathway [64]. Although the discussion between ART-treated HIV-1-infected individuals and elite controllers is modest in the current study, several findings (Figure 1b, Figure 2a and Figure 3b–e) suggest the different propensities of enriched immunologic signatures between these two groups of people living with HIV-1. Given that the specific phenotype of elite control is most likely beneficial to host factors [65], further investigation of the functional interplay between elite controllers and the genome property could be expected to shed some light on the molecular basis of elite control.

At present, transcriptional datasets that correspond to the input gene lists used in this study are not yet available. It is presently unclear whether the transcriptional status of a gene may also be involved in the interaction between HIV-1 integration sites and the host functional genome property. Further orthogonal approaches and additional transcriptomic datasets on corresponding patient cells will be requisites to clarify whether integration site frequency is also dependent on gene expression levels. Another critical issue is that it is not clear whether integration sites of proviruses retrieved by different methods (FLIP-seq [66] and MIP-seq [67] used by Jiang et al. (2020) [6]; PRIP-seq used by Einkauf et al. (2022) [5]) may cause any bias while comparing results between HIV-1-infected individuals and elite controllers. In the future, experiments to retrieve (intact) provirus integration sites from both types of patients in parallel will be required to compare their differences systematically. Nevertheless, the observations in this study allow for proposing a new avenue of research on HIV-1 integration and the host functional genome property alongside HIV-1 infections associated with ART.

## 5. Conclusions

The experimental findings from Einkauf et al. (2022) [5] and Jiang et al. (2020) [6] strongly suggest the presence of immune selection pressure during ART in HIV-1-infected individuals and a long period of elite control, respectively. In this study, I performed the over-representation analysis on provirus-targeted genes from these two studies and demonstrated that unique immunologic signatures were enriched in pretreatment HIV-1-infected individuals, HIV-1-infected individuals subjected to a short period of ART, those subjected to a long period of ART, and elite controllers (Figure 1b). Furthermore, I observed that unique immunologic signatures associated with different combinations of immune cell types (Figure 1e,f) and proinflammatory soluble factors (Figure 2a) would be required to fulfill the host immune response alongside HIV-1 infections associated with ART and a consecutive period of elite control. KEGG pathways enriched by the provirus-targeted genes associated with the signature of proinflammatory soluble factors were frequently related to “cancer specific types”, “immune system”, “infectious disease viral”, and “signal transduction” (Figure 3). Most importantly, this study has shown that immunologic signatures mainly resulted from provirus-targeted genes that interact with HIV-1 (Figure 4). All these findings led us to hypothesize that host genes harboring HIV-1 integration may form different genomic communities, which may define specific immune cell types and proinflammatory soluble factors to support host immunity alongside HIV-1 infections associated with ART.

## Figures and Tables

**Figure 2 vaccines-11-00402-f002:**
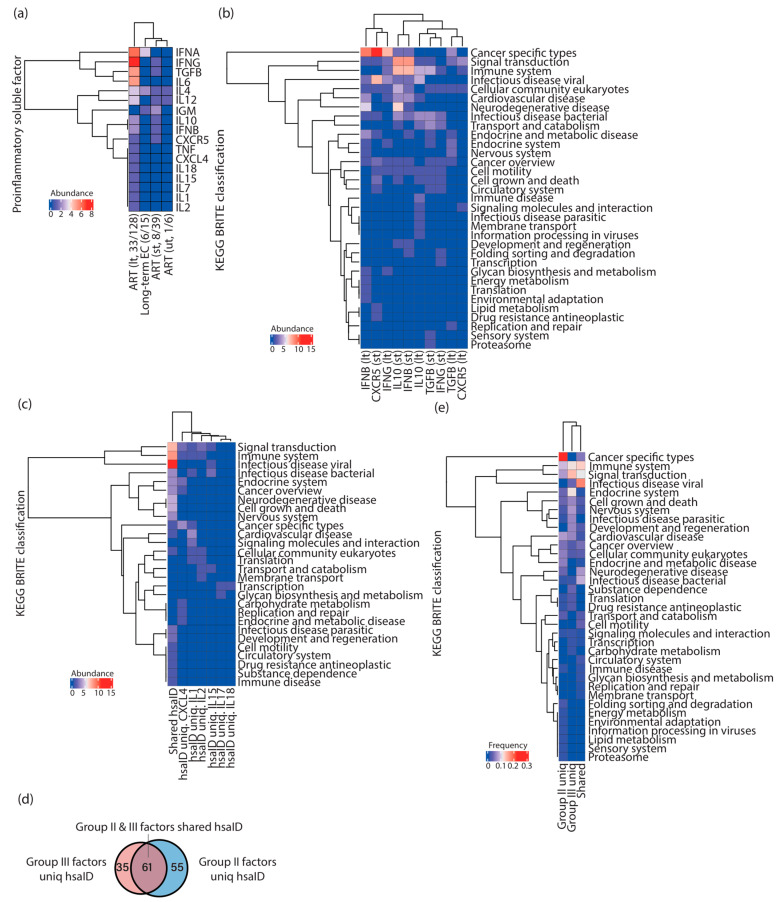
Different compositions of proinflammatory soluble factor signatures were present alongside HIV-1 infections associated with ART. (**a**) Cluster heatmap representing the abundance of proinflammatory soluble factors present in unique immunologic signatures enriched in HIV-1-infected individuals and elite controllers. Parentheses placed under the column indicate the number of proinflammatory soluble factors from the total signatures. (**b**) Cluster heatmap representing the abundance of each KEGG BRITE classification harboring enriched pathways related to group II factors in HIV-1-infected individuals after a short (st) and a long (lt) period of ART. (**c**) Cluster heatmap representing the abundance of each KEGG BRITE classification harboring enriched pathways related to group III factors in HIV-1-infected individuals after a long period of ART. Enriched pathways that were found in all group III factors (so-called “Shared hsaID”) were separated from other enriched pathways unique to individual group III factors to create the cluster heatmap. (**d**) Venn diagram showing the overlap of enriched KEGG pathways represented by hsaID between group II and group III factors. (**e**) Cluster heatmap representing the frequency of KEGG BRITE classifications harboring enriched pathways between group II and group III factors. ART; antiretroviral therapy. EC; elite controllers. “hsaID”; Homo sapiens (human) ID.

**Figure 3 vaccines-11-00402-f003:**
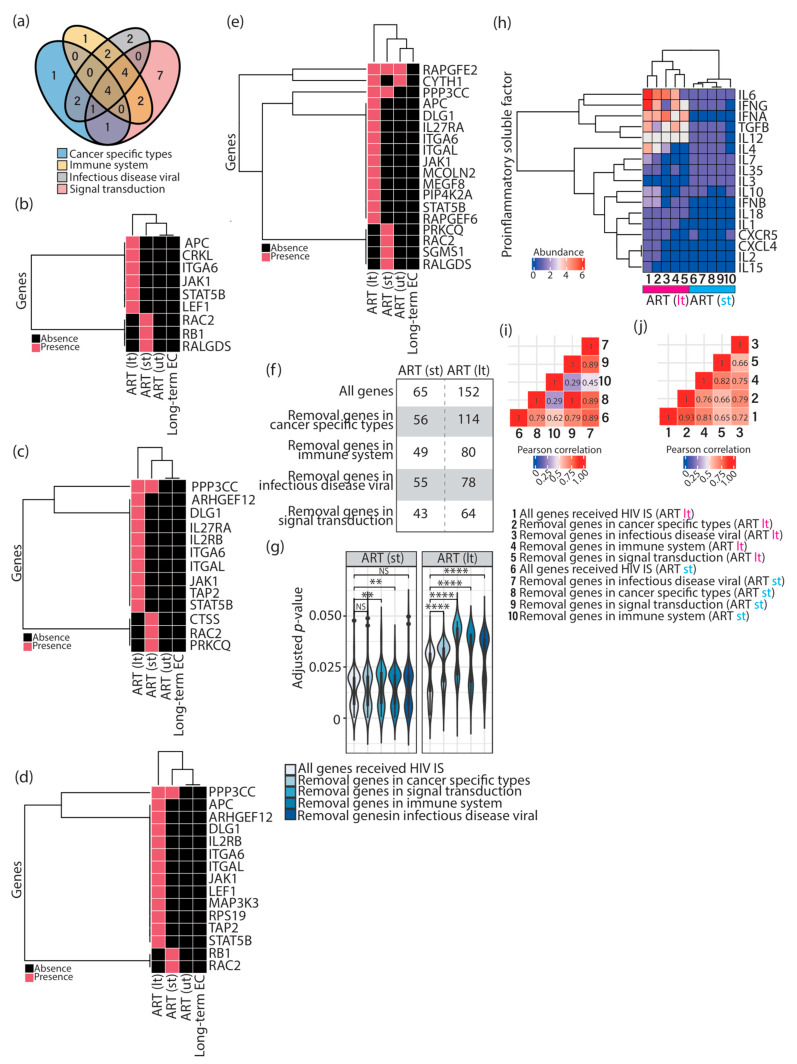
Impact of enriched KEGG pathways involved in “cancer specific types”, “immune system”, “signal transduction”, and “infectious disease viral” in HIV-1-infected individuals subjected to ART. (**a**) Venn diagram representing the overlap of the genes present in enriched KEGG pathways involved in the four KEGG BRITE classifications. (**b**–**e**) Cluster heatmap representing the presence of the genes in enriched KEGG pathways related to “cancer specific types” (**b**), “immune system” (**c**), “infectious disease viral” (**d**), and “signal transduction” (**e**) in HIV-1-infected individuals before ART (ut) and after a short (st) or a long period (lt) of ART, and in elite controllers. (**f**) Numbers of immunologic signatures enriched by either a complete list of input genes (all genes) or a gene list with the removal of the genes in the four KEGG BRITE classifications in HIV-1-infected individuals after a short (st) or a long period (lt) of ART. (**g**) Violin plots representing the distribution of adjusted *p*-values corresponding to immunologic signatures enriched by either a complete list of the genes (all genes) or a gene list with the removal of the genes in the four KEGG BRITE classifications. Statistical significance was calculated using the Wilcoxon test in R with default options. (**h**) Cluster heatmap representing the abundance of proinflammatory soluble factor signatures enriched by either a complete list of the genes (all genes, column 1 and column 6) or a gene list with the removal of the genes in the four KEGG BRITE classifications in HIV-1-infected individuals after a short (st, columns 7–10) or a long period (lt, columns 2–5) of ART. (**i**,**j**) Correlation plots representing the Pearson correlation calculated based on the abundance of proinflammatory soluble factors in HIV-1-infected individuals after a short (**i**) or a long period (**j**) of ART. NS > 0.05, ** *p* ≤ 0.01, **** *p* ≤ 0.0001. ART; antiretroviral therapy. EC; elite controllers. HIV IS; HIV integration sites.

**Figure 4 vaccines-11-00402-f004:**
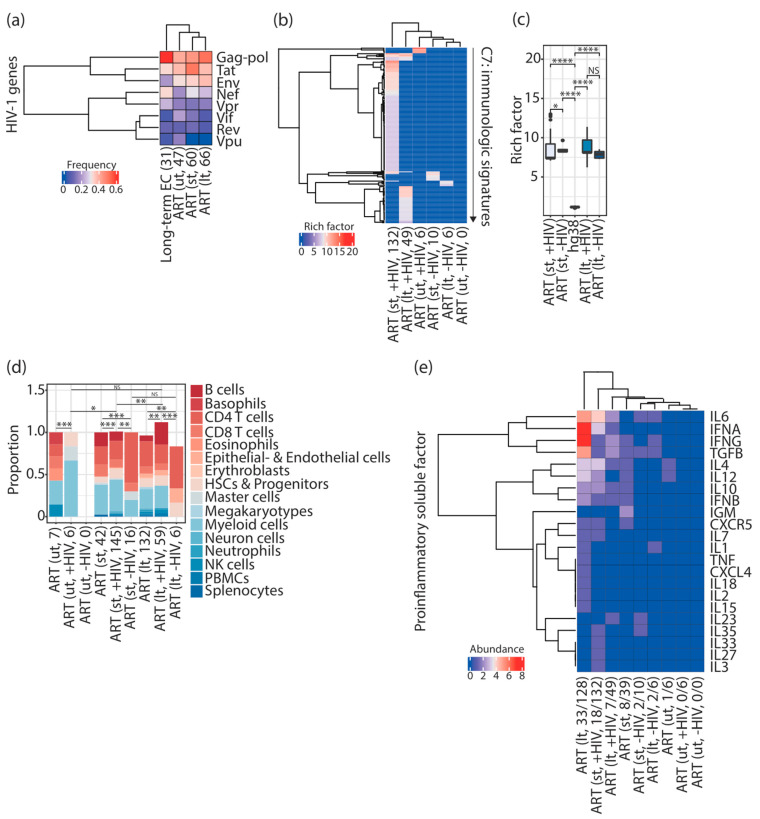
HIV-1-targeted gene sets that interact with HIV-1 contributed to enriching immunologic signatures alongside HIV-1 infections associated with ART. (**a**) Cluster heatmap representing the frequency of provirus-targeted genes interacting with HIV-1. Parentheses placed under the column indicate the total number of provirus-targeted genes reported to interact with HIV-1. (**b**) Cluster heatmap representing enriched immunologic signatures in HIV-1-infected individuals and elite controllers. Inside the parentheses, +HIV and −HIV indicate provirus-targeted genes reported to interact with HIV-1 or not, respectively. The number indicates the number of enriched immunologic signatures. The color scale represents the intensity of the enrichment represented by rich factors. (**c**) Box plots representing the enrichment of immunologic signatures represented by rich factors. Commands executed to calculate rich factors are described in the Materials and Methods. Statistical significance was calculated using the Wilcoxon test in R with default options. (**d**) Stacked bar chart representing the proportion of immune cell types present in enriched immunologic signatures in HIV-1-infected individuals. Statistical significance was calculated using Pearson’s chi-square test in R with default options. Inside the parentheses, +HIV and −HIV indicate provirus-targeted genes reported to interact with HIV-1 or not, respectively. The number indicates the number of counted immune cell types in enriched immunologic signatures. (**e**) Cluster heatmap representing the abundance of proinflammatory soluble factors present in enriched immunologic signatures in HIV-1-infected individuals. Inside the parentheses, +HIV and −HIV indicate provirus-targeted genes reported to interact with HIV-1 or not, respectively. The number indicates the number of proinflammatory soluble factors from the total number of enriched immunologic signatures. NS > 0.05, * *p* ≤ 0.05, ** *p* ≤ 0.01, *** *p* ≤ 0.001, **** *p* ≤ 0.0001. “ut”; pretreatment. “st”; short period of ART. “lt”; long period of ART. ART; antiretroviral therapy. EC; elite controllers. HSCs; hematopoietic stem cells. NK cells; natural killer cells. PBMCs; peripheral blood mononuclear cells.

**Table 1 vaccines-11-00402-t001:** Input gene list.

Pretreatment (ut)	Short Period of ART (st)	Long Period of ART (lt)	Elite Controllers
ACOX1	AAK1	ABCC1	ABCA11P
ACSF3	AC004791.2	ACTR3	AGPAT3
ADARB1	ACAP2	ADGRE5	ANKRD11
ADTRP	ADD3	AHR	APBA2
AIM1	AJ003147.9	ANKRD11	ARHGAP17
ANKLE2	ANKRD11	APC	ARHGAP31
ANKRD31	ARHGAP24	ARHGEF12	ARHGEF18
ARHGAP35	ARIH2	ARHGEF3	ASXL1
ARID4B	ASAP1	ARID1A	ATR
ASNA1	ASH1L	ARNT	BACH2
BRD1	ATXN2L	ASH1L	CEACAM21
BUB1B	BOLA2	ASPSCR1	COX10-AS1
C10orf76	BRD4	ATN1	CREBBP
CCDC26	C16orf52	AUH	DCLRE1C
CCDC57	C17orf62	B3GALT4	DECR1
CD58	C4B	BACH2	DHTKD1
CEP85	C6orf89	BCL7C	DIP2A
CHAF1A	CHD6	BRD4	DIP2A-IT1
CLIP1	CLN5	C16orf72	DLGAP1
CNN2	CLUAP1	C4orf22	ELMO1
CNOT1	CNOT6L	CAMK2D	ENSG00000231731
CNOT4	CNTRL	CCDC30	ENSG00000260989
CRTC3	COL5A1	CCDC57	ENSG00000265987
CSNK1G1	COPB1	CD84	ENSG00000267179
CSNK1G2	CPNE1	CENPC	ENSG00000267552
CTCF	CREBBP	CHMP4B	ENSG00000281016
CTSA	CTSS	CLIP4	ENSG00000283515
CYTH1	CYLD	CNKSR2	ENSG00000285816
DAB1	DARS-AS1	COTL1	ENSG00000288715
DAZAP1	DGKA	CRKL	ENTR1
DNM2	DIAPH1	CUL5	ERC1
EHMT1	DNAJC18	CYTH1	EVL
ETFA	DPF2	DAP3	FCHSD2
FANCA	DYNC1H1	DAPK2	GRB2
FBXL12	EHD4	DAZAP1	HLA-DQA1
FGD3	EP400	DCLRE1C	HSF5
FOXK2	ERCC6L2	DCUN1D1	IFT140
FRMD4B	FAM102A	DLG1	IWS1
FUS	FAM102B	DNAH1	JMJD1C
GBF1	FAM134B	DNAJC18	KANSL2
GRSF1	FANCA	DNMT1	KDM2A
HSPBAP1	FBXL20	DPP8	KIAA0319L
IARS	FGD3	E2F3	KMT2C
IFNAR1	FOXK2	EED	LHPP
IGSF6	FUT8	EIF4G3	LINC00298
IL12RB1	GIMAP8	EPB41	LMTK2
IP6K1	GPR89A	ERCC6L2	MGMT
KIAA2026	GRB2	ERMP1	MYDGF
KIF21A	GTF2E2	ESCO1	MYO1F
KMT2A	HIST1H2AC	EVI5	NELL2
LINC	HMGCS1	EXOC7	NF1
LINC00499	HN1L	FAM13A	NPHP3
LINC00536	HNRNPC	FAM214A	NPHP3-ACAD11
MACF1	HTT	FNBP1	NUMA1
MALAT1	IKZF3	FRY	P4HB
METTL15	IRF2	GATAD2A	PACS1
METTL8	IRGQ	GGA3	PARP4
MFSD12	ITPKB	GON4L	PCNT
MROH1	ITPR3	GRAP2	PIK3IP1
MTG1	KIAA0753	HBS1L	POLA2
NAA38	KXD1	HECTD4	PPM1G
NCOA7	LCOR	HFE	PPP4R3A
NCOR1	LINC00869	HIVEP3	PRDM2
NLRC5	MACF1	IL27RA	RAB14
NPLOC4	MALT1	IL2RB	RABEP1
NT5C2	MAN1A2	IQGAP1	RAD54L2
NUDT3	MAPK14	IQGAP2	RAI1
NUP98	MARS	ITGA6	RAP1GDS1
PARPBP	MAST2	ITGAL	RNF157
PBX3	MCF2L2	JAK1	RPL15P4
PDK3	MDS2	KANSL3	RPTOR
PDXDC2P	MED15	KDM2A	SMG7
PGM2L1	METTL15	KDM3A	SPEN
PPFIA1	MKL1	KIAA0430	SPPL3
PPP2CA	MORC3	KIAA0753	STK38
PRDM2	MRPS6	KIF13B	STX16
PRKAG1	MYH9	LDLRAD4	STX16-NPEPL1
PRRC1	NACC2	LEF1	TBCD
PSMB2	NCAPD2	LPP	TMEM245
PTPN1	NCBP3	LSM14A	TNRC6A
RAPGEF2	NDC80	LUZP1	TPTE2P6
RBX1	NDUFA6	MALAT1	TRAPPC14
RN7SKP127	NDUFAF2	MAP3K3	TRIM37
RNF214	NMD3	MCOLN2	TRIT1
RNF216	NPEPPS	MEGF8	UBE2O
RNF34	NR2C2	MICB	UBE3C
RPTOR	OS9	MIR3681HG	UBR2
RRN3	PACS1	MROH1	UTP20
RUNX2	PIBF1	MYCBP2	ZMYM4
SARS	PIK3R1	NDUFA6	ZNF160
SCMH1	PLEC	NFKBIL1	ZNF180
SECISBP2L	PPP3CC	NSD1	ZNF225
SEPT9	PPP6R3	NUDT3	ZNF274
SETD3	PRKCA	NUDT4	ZNF350
SMARCC1	PRKCQ	NVL	ZNF350-AS1
SPG11	PRR19	OGDH	ZNF407
STAU1	PSME4	PATL1	ZNF430
TAF1	PSMF1	PBRM1	ZNF544
TBC1D4	PTPN4	PCBP2	ZNF607
TGFB1	QRICH1	PHACTR4	ZNF700
THRAP3	RAB11FIP3	PHC1	ZNF720
TMEM87A	RAC2	PHF12	ZNF721
UBAP1	RAD51	PIP4K2A	ZSCAN16
UBE2D2	RALGDS	PITPNC1	ZSCAN16-AS1
UBE2G1	RAPGEF2	PLA2G16	
UMAD1	RB1	PPP3CC	
UNC13B	RBL1	PPP6R1	
USP3	RCOR1	PRR19	
VAV1	REV1	PUM1	
VDR	RNF157	PXK	
VPS13B	RPRD2	RAD18	
WARS2	RPTOR	RANBP9	
ZAN	SACM1L	RAPGEF2	
ZBED5	SAE1	RAPGEF6	
ZNF226	SCYL1	RBM15	
ZNF557	SGMS1	RFFL	
ZNF701	SKAP1	RIMKLB	
ZNF737	SMCHD1	RPS19	
ZNF850	SMG6	RPS6KA3	
ZNF91	SNX25	RPTOR	
ZSCAN22	SPECC1L	RSRC2	
	STK11	SAE1	
	TARBP1	SCNN1A	
	TATDN2	SERPINH1	
	TBK1	SETX	
	TEX2	SF3B2	
	TMEM65	SFMBT1	
	TNRC6C	SHCBP1	
	TOM1L2	SLC23A2	
	TPM3	SLC25A40	
	TRAF3	SLC30A7	
	TRPC4AP	SLC35A3	
	TUBA8	SLC44A2	
	TYW1	SLC9A6	
	UBOX5	SLX4IP	
	USP25	SNTB2	
	USP49	SPEN	
	USP6NL	ST13	
	USP9Y	ST7	
	VCL	STAT5B	
	VRK1	SUMO2	
	WASH1	TAB1	
	WBP1L	TAF2	
	WHSC1L1	TANGO6	
	WNK1	TAP2	
	ZBED4	TBC1D2B	
	ZC3H18	TCF3	
	ZNF564	TMBIM6	
	ZNF609	TNFSF12	
		TNRC6B	
		TNRC6C	
		TOM1L2	
		TP53TG5	
		TRAT1	
		TSEN54	
		TSG101	
		UBA6	
		UBE2Q2	
		UBTD2	
		UPF1	
		USP34	
		VARS	
		VAV1	
		VPS13D	
		VPS8	
		WASF2	
		XPO6	
		Y_RNA	
		YWHAE	
		ZCCHC11	
		ZCCHC7	
		ZKSCAN8	
		ZNF140	
		ZNF451	
		ZNF529	
		ZNRF1	

A list of the provirus-targeted genes used for over-representation analysis in this work. Provirus integration sites were retrieved from Jiang et al. (2020) [6] for elite controllers and Einkauf et al. (2022) [5] for HIV-1-infected individuals.

**Table 2 vaccines-11-00402-t002:** The number of provirus-targeted genes (from Table 1) associated with signatures of immune cell types and proinflammatory soluble factors.

Patient Sample	Number of Provirus-Targeted Genes (Number of Enriched Immunologic Signatures)	
	Immune Cell Types	Proinflammatory Soluble Factors
Pretreatment (ut)	35 (6)	7 (1)
Short period of ART (st)	101 (37)	37 (7)
Long period of ART (lt)	164 (121)	109 (33)
Elite controllers	48 (14)	25 (6)

## Data Availability

Publicly available datasets were analyzed in this study.

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
