# Peer review of "The Dynamic Linkage between Provirus Integration Sites and the Host Functional Genome Property Alongside HIV-1 Infections Associated with Antiretroviral Therapy"

_vaccines, 2023, doi:10.3390/vaccines11020402_

Round 1
Reviewer 1 Report
The manuscript by Chen attempts to link intact provirus integration sites with the activation of biological pathways, particularly involved in host immunity. This manuscript suffers from a lack of details concerning the patients, the number of intact proviral genomes that were analyzed per patient, and the expression of these immunological factors. I don’t see how the information from this study can be used as the integration sites in any given HIV-1 individual are likely to be different.
Major comments:
1. Line 45-47, In the sentence, “In the continuation of this concept, in this work, I sought any functional linkage between the configuration of the HIV-1 reservoir and host immunity alongside HIV-1 infections. What is meant by the “configuration of the HIV-1 reservoir and host immunity?”
2. lines 66-68. The author should present more information regarding the number of patients analyzed. The author uses data from two previous publications. The study by Einkauf et al. (2022) used 7 elite controllers. The second study by Jiang et al. (2020) used 105 patients for pretreatment, short term, and long-term ART. How many patients were analyzed per group in the present study and how many integrated intact proviruses were analyzed per patient?
3. Lines 106-112: basophils, eosinophils, epithelial, endothelial, erythroblasts, hematopoietic, progenitors, master, megakaryocytes, myeloid, neuron, neutrophils, peripheral, and splenocytes should not be capitalized.
4. Lines 169-170: In the sentence containing “and in elite controllers showed significance compared 169 to controls (Figure 1b)” please define what the controls are.
5. The logic behind examining different immune cell phenotypes escapes me. HIV-1 infects and integrates into only a fraction of these cell types.
6. A major concern with the study is that while the previous papers showed that the proviruses were transcriptionally active, no data was presented that these cells produce infectious virus.
6. The author lists genes on six pages from the four groups analyzed but makes no attempt to analyze them across the four different groups.
7. The author suggests that gene sets harboring intact proviruses may define immune cells and proinflammatory factors associated with ART therapy. While the detection of the transcriptionally active gene is of value unless this is confirmed by analysis of protein levels the results will be equivocal at best.
8. Were the patients all given the same ART therapy regimen? This is not stated in the manuscript and I could not find the details in the original papers that are used as the basis of this study.
Minor comments:
1. Throughout the manuscript “HIV” should be changed to “HIV-1.”
2. line 32: eliminate the comma after factors.
3. line 33: change “this” to “these.”
4. line 34: change “mechanism” to “mechanisms.”
5. line 34: change “initiates” to “begins.”
6. line 37: the sentence, “targeted by HIV-1 in HIV-1 infected individuals” should be changed to “targeted by HIV-1 in infected individuals.”
7. line 80: change “smaller” to “less.”
8. line 156: “8” should be changed to “Eight.”
Reviewer 2 Report
The manuscript quite well reports intact provirus integration sites and the host functional genome property alongside HIV-1 infections associated with antiretroviral therapy. The data and analysis are well done.
There is only a minor suggestion for the author.
Further discussion is required for the observed genes such as immunologic signatures and cytokines genes which could enrich references, meanwhile supply more information for readers.
Round 2
Reviewer 1 Report
I have reviewed the resubmission of the above manuscript. The author has addressed the concerns I have raised in original submission.
